

# ERpred: a web server for the prediction of subtype-specific estrogen receptor antagonists

Nalini Schaduangrat, Aijaz Ahmad Malik and Chanin Nantasenamat

Center of Data Mining and Biomedical Informatics, Faculty of Medical Technology, Mahidol University, Bangkok, Thailand

## ABSTRACT

Estrogen receptors alpha and beta (ERα and ERβ) are responsible for breast cancer metastasis through their involvement of clinical outcomes. Estradiol and hormone replacement therapy targets both ERs, but this often leads to an increased risk of breast and endometrial cancers as well as thromboembolism. A major challenge is posed for the development of compounds possessing ER subtype specificity. Herein, we present a large-scale classification structure-activity relationship (CSAR) study of inhibitors from the ChEMBL database which consisted of an initial set of 11,618 compounds for ERα and 7,810 compounds for ERβ. The $IC_{50}$ was selected as the bioactivity unit for further investigation and after the data curation process, this led to a final data set of 1,593 and 1,281 compounds for ERα and ERβ, respectively. We employed the random forest (RF) algorithm for model building and of the 12 fingerprint types, models built using the PubChem fingerprint was the most robust (Ac of 94.65% and 92.25% and Matthews correlation coefficient (MCC) of 89% and 76% for ERα and ERβ, respectively) and therefore selected for feature interpretation. Results indicated the importance of features pertaining to aromatic rings, nitrogen-containing functional groups and aliphatic hydrocarbons. Finally, the model was deployed as the publicly available web server called ERpred at http://codes.bio/erpred where users can submit SMILES notation as the input query for prediction of the bioactivity against ERα and ERβ.

## INTRODUCTION

Breast cancer is the most frequently detected cancer amongst women with over 2 million new cases and an estimated 627,000 deaths (15% of all cancer deaths in women) in 2018, according to the WHO (*World Health Organization, 2018*). Furthermore, it is a well-known fact that levels of Estrogen Receptor (ER) impacting breast cancer metastasis are the fundamental and critical determinants of clinical outcomes (*Kammerer et al., 2013*; *Gamucci et al., 2013*). In addition, ER positive breast cancer types exhibit favorable responses to hormone therapy (*Althuis et al., 2004*; *Foulkes, Smith & Reis-Filho, 2010*; *Thrane et al., 2013*), for example tamoxifen (*Ramirez-Ardila et al., 2013*), or aromatase

Corresponding author
Chanin Nantasenamat,
chanin.nan@mahidol.edu

inhibitors (*Hiscox, Davies & Barrett-Lee, 2009*), designed to block aberrant signaling within oncogenic pathways. However, a major obstacle in the case of chemotherapy in ER-positive breast cancers is chemoresistance (*Kim et al., 2010*; *Ji et al., 2019*; *Han et al., 2019*). Therefore, new systemic therapies are urgently needed.

The estrogen receptor is a member of the nuclear receptor family, mainly found in the nucleus but can also be seen in the cytoplasm and mitochondria. ER consists of two main subtypes (i.e., ERα and ERβ) which bind to hormones and trigger the activation or repression of genes (*Brzozowski et al., 1997*). Estrogen signaling is selectively stimulated or inhibited depending on the balance between ERα and ERβ activities in target organs. Both receptor subtypes are expressed in various cells and tissues (i.e., breast, prostate and ovary) as they control various physiological functions of the human body (i.e., reproductive, skeletal, cardiovascular and central nervous systems). The mammary gland, uterus, ovary (thecal cells), bone, male reproductive organs, prostate, liver and adipose tissues are mainly composed of ERα (*Welboren et al., 2009*). In contrast, ERβ is found mainly in the prostate, bladder, ovary (granulosa cells), colon, adipose tissue, and the immune system (*Weiser, Foradori & Handa, 2008*). Common physiological roles for ERα and ERβ includes the development and function of ovaries and the protection of the cardiovascular system (*Paterni et al., 2014*). A more prominent role is exerted on the mammary gland and uterus as well as on the homeostasis of the skeletal system and the regulation of metabolism by ERα. ERβ on the other hand, exerts a more powerful effect on the central nervous and immune systems (*Paterni et al., 2014*). Furthermore, the β subtype is shown to generally counteract the hyperproliferation of ERα-promoted cells in tissues such as the breast and uterus (*Heldring et al., 2007*).

Estradiol and hormone replacement therapy targets both ERs, but this often leads to an increased risk of breast and endometrial cancers as well as thromboembolism. Selective estrogen receptor modulators (SERMs) are the most common drug group used in ER-positive breast cancer treatment with tamoxifen as the first line agent used to block mitogenic effects of estrogen at all stages of breast cancer, particularly in pre- and post-menopausal patients (*Abdulkareem & Zurmi, 2012*). In addition, Fulvestrant, the main drug in the group of Selective estrogen receptor downregulators (SERDs) is used as an alternative in tamoxifen resistant breast cancers and acts by disrupting the ER receptor and blocking ER dimerization which in turn inhibits estrogen signaling via ER down-regulation (*Osborne, Wakeling & Nicholson, 2004*). However, the effectiveness of fulvestrant is decreased by acquired resistance whereby a response to therapy is not seen in most ER-positive breast cancer patients (*Cook, Shajahan & Clarke, 2011*). Furthermore, through mechanisms distinct from ER subtype selective binding, SERMs target both receptor subtypes even though they display tissue-selective agonist/antagonist activities (*Paterni et al., 2014*). An ideal SERM would thus possess antagonist activity in the mammary gland and uterus and antagonist activity in other tissues such as those pertaining to the skeletal, cardiovascular or the central nervous systems (*Jordan, 2001*). Alternatively, based on the distribution and levels of the two ER subtypes in the various tissues mentioned above, subtype-selective ligands could be used to elicit beneficial estrogen-like activities and reduce side effects. In this regard, there appears to be particular

promise for the use of subtype-selective agonists/antagonists (*Kumar et al., 2011*). However, the challenge for developing ER subtype specific compounds remains elusive.

The ER structure contains a globular ligand binding domain (LBD) harboring a hormone-binding site, a homo- or heterodimerization interface, and coregulator (activator and repressor) interaction sites (*Kumar et al., 2011*). The amino acid sequence of ERα and ERβ displays a 55% sequence identity in their respective LBDs, which represents a significant difference (*Kerdivel, Habauzit & Pakdel, 2013*). The LBD of both ER subtypes are comprised of 12 α-helices (H1–H12) arranged in a three-layered sandwich topology (shown in Fig. 1) with a central core layer of three helices (H5/6, H9 and H10) sandwiched between two additional layers of helices (H1–4 and H7, H8, H11) in an anti-parallel formation. The remaining secondary structural elements, a small two-stranded antiparallel β-sheet (S1 and S2) and H12, are located at the narrow end of the ligand-binding portion of the molecule (*Brzozowski et al., 1997*). Upon binding to its natural ligand (i.e. 17β-estradiol or E2), a deep hydrophobic environment is formed through an ellipsoidal cavity (*Kumar et al., 2011*). In addition, the hydroxyl groups form the A and D rings of E2, which are comprised of hydrogen bonded residues in H3, H5 and H11. These hydrogen bonds play crucial roles in the orientation of the steroid ligand (*Brzozowski et al., 1997*). Upon agonist binding, both ER subtypes orient the H12 helix to create a hydrophobic pocket for interaction with the LXXLL motif of coactivators (*Leung et al., 2006*). This binding allows for conformational changes to occur which in turn activates or inhibits responsive genes (*Arao et al., 2013*). However, upon antagonist binding, the alignment of H12 over the binding cavity is prevented and thus, no co-factor binding and downstream gene activation takes place. This antagonist-induced repositioning of H12 is considered to be a crucial step in the prevention of ER activation (*Dahlman-Wright et al., 2006*). Structural analysis of various antagonist-ER complexes has revealed that they usually contain a bulky side chain that cannot be contained in the ligand binding pocket. The protrusion of these side chain disrupts the binding and subsequent activation of H12 and coactivators (*Shiau et al., 2002*). Owing to the very subtle differences in the LBDs of both ER subtypes, the design of a subtype-specific antagonist is a challenge. Nevertheless, major advances over the past two decades in the fields of structural biology pertaining to ERs have shed light on the plasticity and binding modes of both ER subtypes (*Bafna et al., 2020*; *Brzozowski et al., 1997*; *Pavlin et al., 2018*; *Pang et al., 2018*; *Shiau et al., 2002*).

The discovery of novel drugs is an expensive and time-consuming endeavor. Nowadays, the use of computational methods is increasingly playing important and integral roles as part of the drug discovery process. Quantitative structure-activity relationship (QSAR) is a ligand-based approach that allows elucidation on the prediction and rationalization of the investigated biological activity as a function of computed molecular descriptors that describes the unique physicochemical properties of molecules (*Nantasenamat, Isarankura-Na-Ayudhya & Prachayasittikul, 2010*; *Cherkasov et al., 2014*). QSAR has been successfully applied to model a wide range of bioactivities

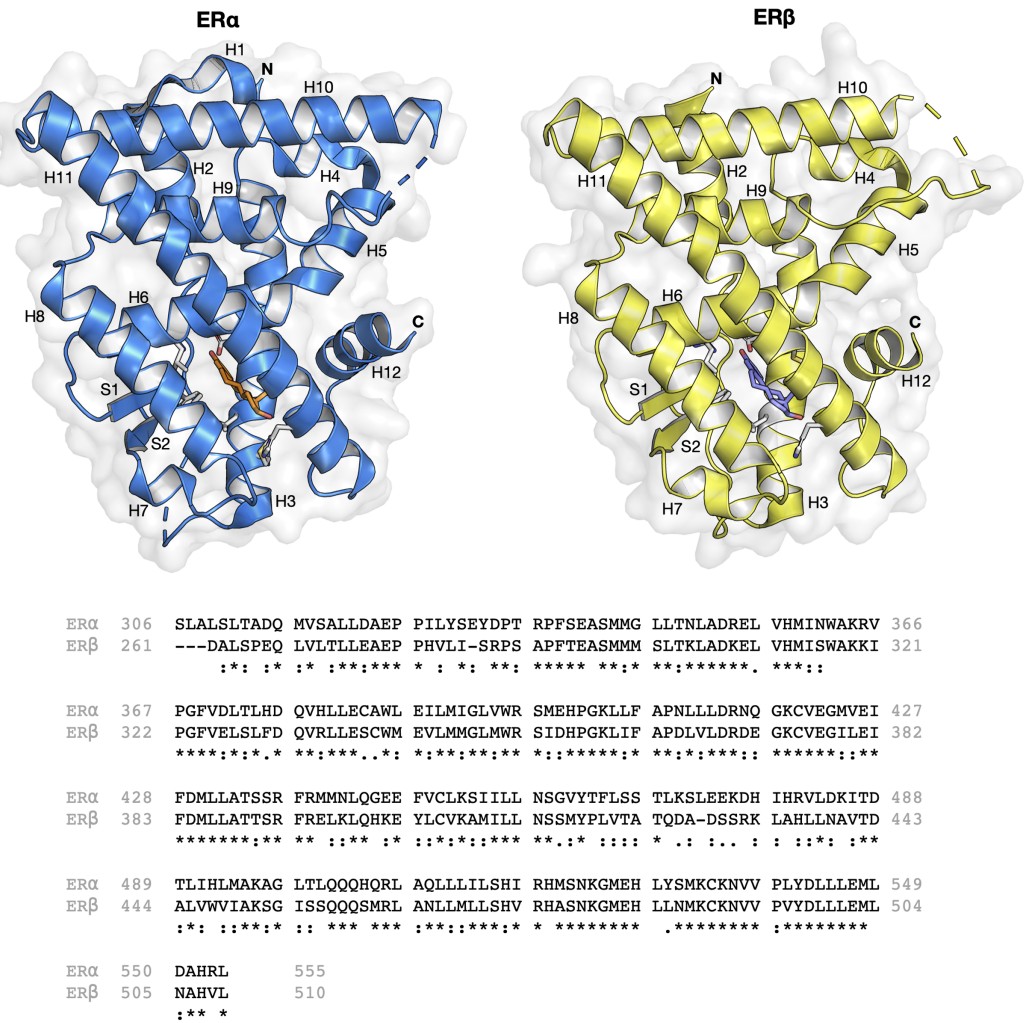

**Figure 1 Protein structure and sequence alignment of ER subtypes.** ERα and ERβ are displayed in blue and yellow colors, respectively. Secondary structure elements consisting of 12 helices and two strands are shown. Sequence alignment of the ligand binding domain of the two ER subtypes were performed in Clustal Omega.

and chemical properties. Such models are also useful for rationalizing the importance and contributions of molecular features on investigated activities/properties. Subsequently, crucial information pertaining to ER binding affinity coupled with structure-binding and structure-activity relationship data, have led to the formulation of reliable ERα models (*Anstead, Carlson & Katzenellenbogen, 1997*; *Serafimova et al., 2007*; *Xiang et al., 2009*; *Toropov et al., 2012*; *Chang et al., 2013*; *Ribay et al., 2016*; *Suvannang et al., 2018*; *Lee & Barron, 2017*; *Pavlin et al., 2018*; *Pang et al., 2018*; *Balabin & Judson, 2018*; *Cotterill et al., 2019*; *Bafna et al., 2020*). However, not much research has been conducted regarding the binding specificity towards ERβ (*Manas et al., 2004*; *Coriano et al., 2018*). Thus, the aim of this study is to build classification models able to (i) distinguish active from inactive compounds for both ERα and ERβ, and (ii) build a web server for discriminating compounds for estrogen receptor α and β with selectivity.

## MATERIALS AND METHODS

### Data compilation and curation

Two data sets of inhibitors against ERα and ERβ were compiled from the ChEMBL database, version 25 (*Gaulton et al., 2017*), which consisted of an initial set of 11,618 compounds for ERα and 7,810 compounds for ERβ. The $IC_{50}$ was selected as the bioactivity unit for further investigation and after the data curation process, this led to a final data set of 1,593 and 1,281 compounds for ERα and ERβ, respectively. As this study sets out to develop a classification model for both ERα and β we therefore, defined thresholds of <1 and >10 μM (corresponding to $pIC_{50}$ values of 6 and 5, respectively) for distinguishing actives from inactives, respectively. Moreover, the intermediate biological activity with $IC_{50}$ values ranging between 1 and 10 μM were not selected for this study. A final set of non-redundant and curated compounds consisting of 1,194 and 997 inhibitors were obtained for ERα and ERβ, respectively.

### Molecular descriptors

Fingerprint descriptors for compounds in the data sets were computed using the PaDEL-descriptor software (*Yap, 2011*). SMILES notation was used for the calculation of molecular descriptors. Structures were pre-processed so as to remove salt and standardize tautomers using the built-in function of the PADEL-descriptor software. In general, molecular descriptors are important for QSAR studies as they characterize molecular properties and chemical structure information in quantitative or qualitative forms. As previously described in *Malik et al. (2020)* 12 molecular fingerprints belonging to 9 classes consisting of AtomPairs 2D, CDK fingerprinter, CDK extended, CDK graph only, E-state, Klekota–Roth, MACCS, PubChem and Substructure were used for describing the chemical structures. Furthermore, Substructure, Klekota–Roth fingerprint and 2D atom pairs consisted of two versions: (1) 1 or 0 denotes the presence or absence of the descriptor and the (2) representing the frequency value of the descriptor with a count version.

### Data filtering

In order to remove inherent complexity and bias that may be introduced to the model building process, constant and near constant variables were removed to select the fingerprint descriptor sets. Particularly, near constants were identified using a standard deviation (SD) threshold of 0.1 whereby variables with SD values less than 0.1 were selected for further analysis.

### Data splitting

The Kennard–Stone algorithm (*Kennard & Stone, 1969*) was applied for splitting the data into an 80/20 split where, the internal set comprised of 80% of the entire data set while the external set consisted of the remaining 20%. The internal set is used as the training set and also subjected to 5-fold CV. The external set is used as the testing set whereby the trained model will be applied to this data set to make predictions and thus determine the model's robustness. For the internal set of ER alpha (comprising 528 compounds), active

compounds accounted for 53.59% while inactive compounds are comprised of 46.41%. Similarly, for the external set (comprising of 131 compounds), active and inactive compounds were split into sets of 53.44% and 46.56%, respectively. For ERβ, the percentage of active and inactive in the internal data set (composed of 572 compounds) was 78.15% and 21.85%, respectively. Similarly, in the external data set (composed of 142 compounds), the percentage for active and inactive was 78.12% and 21.88%, respectively.

## Statistical analysis

As described in our previous work (*Malik et al., 2020*), trends in individual descriptors of active and inactive compounds were determined through 6 common descriptive statistical parameters, encompassing the minimum (Min), first quartile (Q1), median, mean, third quartile (Q3) and maximum (Max) parameters. The outcome was visualized in the form of a box plot using ggplot2, a package of the R program. In addition, the Mann–Whitney *U* test (also known as the Wilcoxon Rank Sum test) was conducted to determine the statistical significance in terms of the *p*-value.

## Multivariate analysis

Random forest (RF) is used for the building of classification models owing to its robust model performance and interpretability. RF was successfully used in our recent work for modeling the bioactivity of hepatitis C virus inhibitors (*Malik et al., 2020*). In essence, RF is an ensemble classifier that employs an *N* number of decision trees (specified by the *ntree* parameter) to learn the inherent patterns from the input data (*Breiman, 2001*; *Breiman et al., 1984*). In this study, a five-fold cross-validation (5-fold CV) procedure was applied for tuning the *ntree* parameter (100, 1,000, 100) and the *mtry* parameter (5, 30, 5) via the use of the tuneRF function from the *randomForest* package (*Liaw & Wiener, 2002*). In order to provide a better understanding of the biochemical activity of the inhibitors, feature selection was estimated using the built-in importance estimator of the RF model. The mean decrease of the Gini index (MDGI) was utilized to estimate the important descriptors (*Weidlich & Filippov, 2016*). Descriptors affording the largest value of MDGI represents the most important features as that descriptor contributes most significantly to the model performance.

## Model validation

Parameters commonly used for evaluating the model performance of binary classification problems are typically based on true positives (TP), true negatives (TN), false positives (FP) and false negatives (FN). Particularly, the fitness of the model was assessed using various statistical parameters including the overall prediction accuracy (Ac), sensitivity (Sn), specificity (Sp) and Matthews correlation coefficient (MCC) (*Song & Tang, 2004*).

$$Ac = TP + TN/(TP + TN + FP + FN) \times 100 \tag{1}$$

$$Sn = TP/(TP + FN) \times 100 \tag{2}$$

$$Sp = TN/(TN + FP) \times 100 \tag{3}$$

$$MCC = (TP \times TN) - (FP \times FN)/\sqrt{(TP + FP)(TP + FN)(TN + FP)(TN + FN)} \tag{4}$$

where TP, TN, FP and FN represent the instances of true positives, true negatives, false positives and false negatives, respectively.

### Applicability domain analysis

The main purpose of the applicability domain (AD) is to estimate the boundaries within which the model can make reliable and accurate predictions for compounds on the basis of similarity with the compounds on which the model was constructed. The compounds that satisfy the scope of the model are within the AD while the rest are outside the AD. In this study, we used the principal component analysis (PCA) bounding box to assess the AD of compounds from the training (internal) and testing (external) sets.

### Web server development

The predictive model was exported as an RDS file (i.e., the model.RDS file) and subsequently deployed as a web server. Particularly, the web server is coded in R via the use of the Shiny R package (i.e., a web framework for the R environment). Technically, the web server is comprised of two major components: (1) user interface and (2) server, which are saved as ui.R and server.R, respectively. The ui.R file accepts input values (i.e. the SMILES notation of query compounds) and transfers this information to the server.R file where the SMILES notation is submitted to the PaDEL-Descriptor software. After descriptor calculation, the computed descriptors are used as input to the predictive model (i.e., the model is exported as model.RDS file) which will then classify query compounds as either being active or inactive (i.e., the bioactivity class label). Such predicted class labels are then printed out onto the web server whereby users can also download the predicted results as a CSV file.

### Reproducible research

The data and code used in this study are publicly available on GitHub at https://github.com/chaninlab/ERpred/.

## RESULTS AND DISCUSSION

A schematic summary of the workflow employed in this study is shown in Fig. 2. Briefly, we start out by performing a chemical space analysis followed by QSAR model building and an in-depth feature analysis and finally deployed the best predictive model as a public web server.

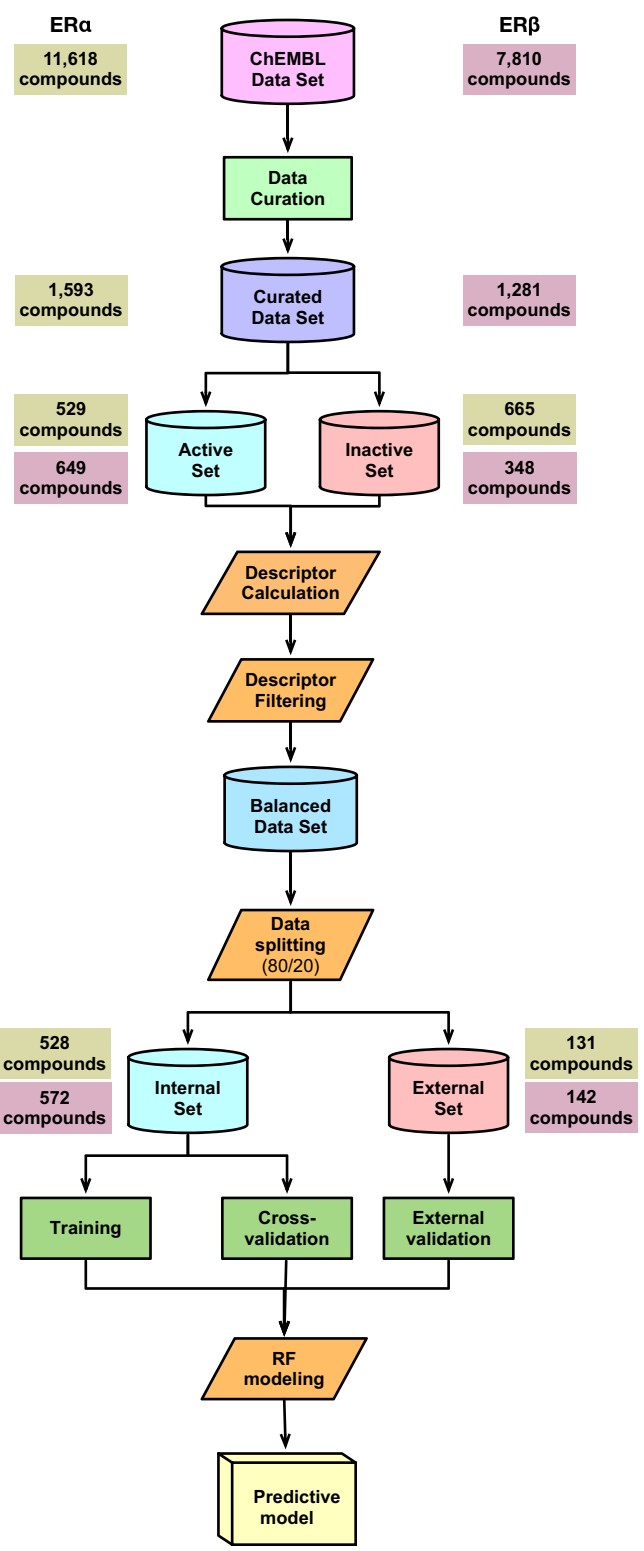

**Figure 2  Schematic representation of the methodological workflow of this study.**

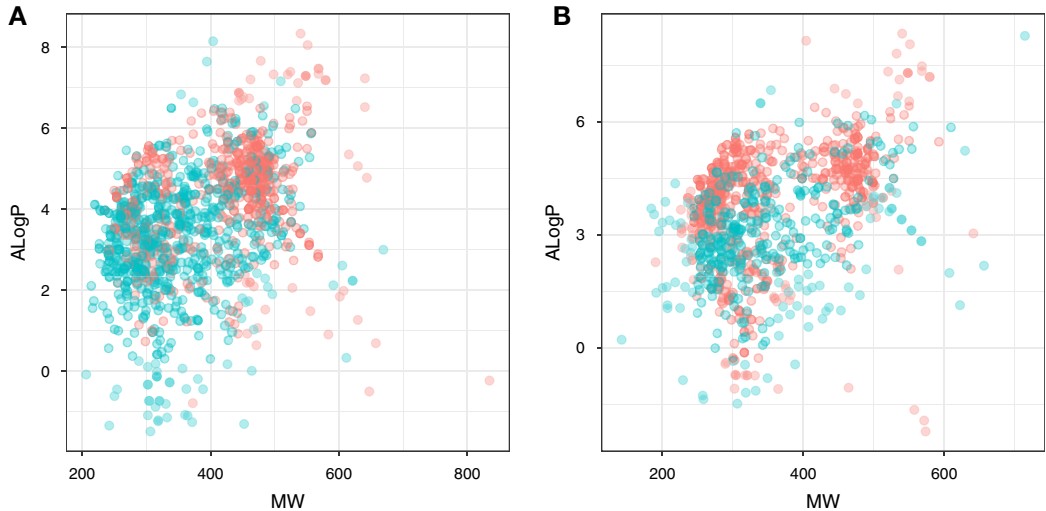

**Figure 3 Plot of MW vs ALogP for compounds in the ERα and ERβ datasets.** The plot allows simple visualization of the chemical space of inhibitors against ERα (A) and ERβ (B). Active and inactive compounds are shown in salmon pink and teal colors, respectively.

## Chemical space analysis

Chemical space analysis is employed to explore the characteristic differences between the active and inactive compounds. The general chemical space was first visualized as a function of the molecular weight (MW) vs the Ghose–Crippen–Viswanadhan octanol-water partition coefficient (ALogP). In addition, the active and inactive compounds were further compared using the Lipinski's rule-of-five (Ro5) descriptors. Briefly, the Ro5 describes the drug likeness of compounds on the basis of their molecular properties namely molecular weight (<500), octanol–water partition coefficient (ALogP < 5), the number of hydrogen bond acceptors (<10) and the number of hydrogen bond donors (<5) (*Lipinski et al., 2001*). Visualization of the MW chemical space as a function of ALogP is shown in Fig. 3. As can be observed for both ERα and ERβ, most of the compounds are clustered within the MW range of 200–500 Da with an ALogP in the range of 1 and 6. In addition, Fig. 4 shows the distribution of active and inactive compounds according to the Ro5 descriptors. It is observed that both ER subtypes contain compounds following the Ro5 criteria such as, MW of less than 500 Da, ALogP value of less than 5 and nHBDon and nHBAcc values of less than 10. It can also be seen that for ERα, some of the active compounds have an ALogP of greater than 5, but the number is very minimal. Furthermore, the results from statistical analysis displays a significant difference between the active and inactive compounds using the Mann–Whitney *U* test. Most of the active compounds (422.68 ± 91.52) were larger (i.e., higher MW) than the inactive compounds (350.35 ± 79.82), which was observed from the mean values of box plots. Similarly, the ALogP values of the active compounds (4.36 ± 1.37) were greater than the inactive compounds (3.17 ± 1.53). However, it was observed that both active and inactive compounds had similar nHBDon values while the active compounds had nHBAcc values that were lower than the inactive compounds. On the other hand, for ERβ, the MW

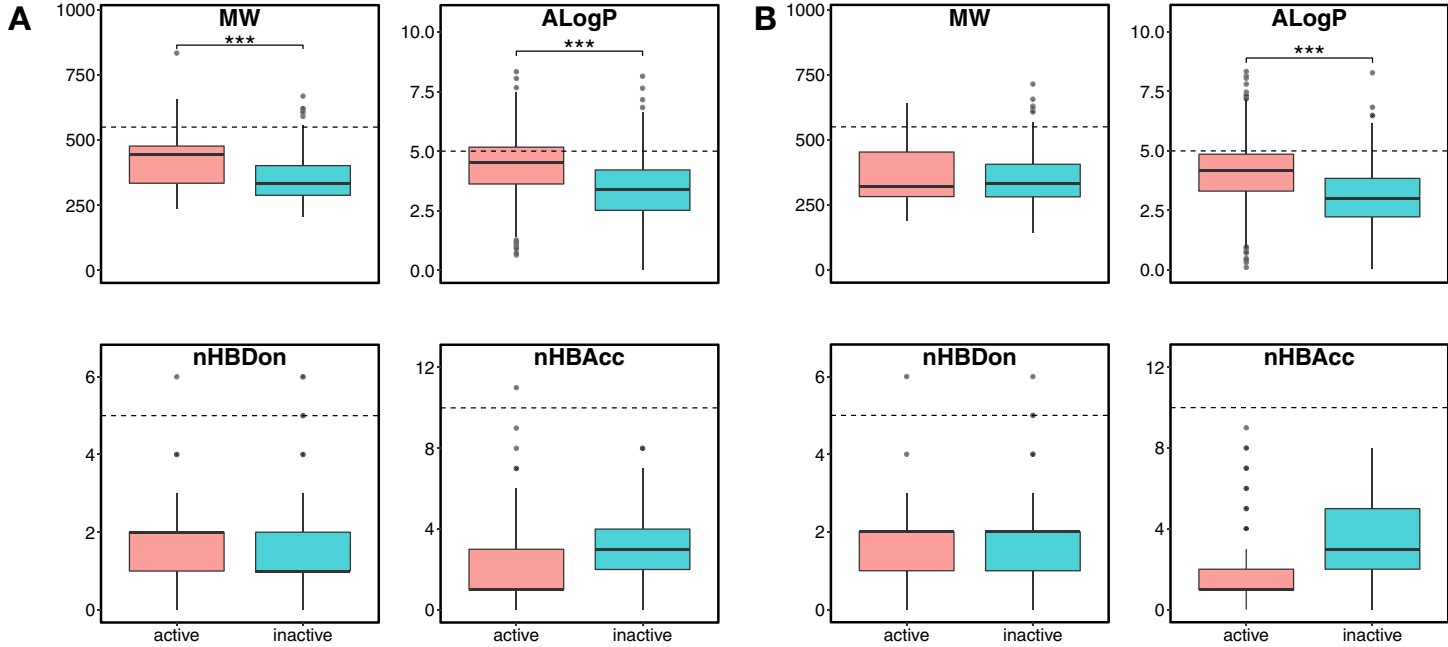

**Figure 4  Box plot of Lipinski's rule-of-five descriptors.** The four rule-of-five descriptors are shown for the ERα (A) and ERβ (B) datasets. Active and inactive compounds are shown in salmon pink and teal colors, respectively.               

between the active (356.94 ± 92.43) and inactive compounds (351.69 ± 94.80) was not statistically significant as determined using the Mann–Whitney U test. Nonetheless, the ALogP was very statistically significant with the active group (3.82 ± 1.6) displaying higher values than the inactive group (2.91 ± 1.5). Similar to the ERα subtype, the nHBDon values of both the active and inactive groups were on par while the nHBAcc for the active compounds was seen to be a lot lower than the inactive compounds.

Moreover, PubChem fingerprints were utilized for estimating the AD of the CSAR model developed herein which were further used as input values for PCA analysis. The resulting PCA scores plot can be seen in Fig. 5. The data set for ERα and ERβ comprised of 1,194 and 997 compounds, respectively were further divided into internal (80%) and external sets (20%) using the Kennard–Stone algorithm. Of note, the training set is composed of the internal set which is utilized to build the model and thus make predictions on the external set. In addition, both the internal and the external sets are also subjected to 5-fold CV. Furthermore, the chemical space distribution as observed from Fig. 5 shows that the external set (i.e., the testing set, represented using blue dots) lies within the boundaries of the internal set (i.e., the training set, represented using red dots). Thus, the AD is well defined for the CSAR model developed herein, as shown through these results.

In order to develop a deeper understanding of the ER chemical space, the active and inactive datasets for both ERα and ERβ were sorted according to their pIC$_{50}$ values. The top 10 active and bottom 10 inactive compounds were taken from each set and applied to the Scaffold Hunter software for further analysis (*Schäfer et al., 2017*). Particularly, the major scaffolds identified were then observed in further detail in terms of

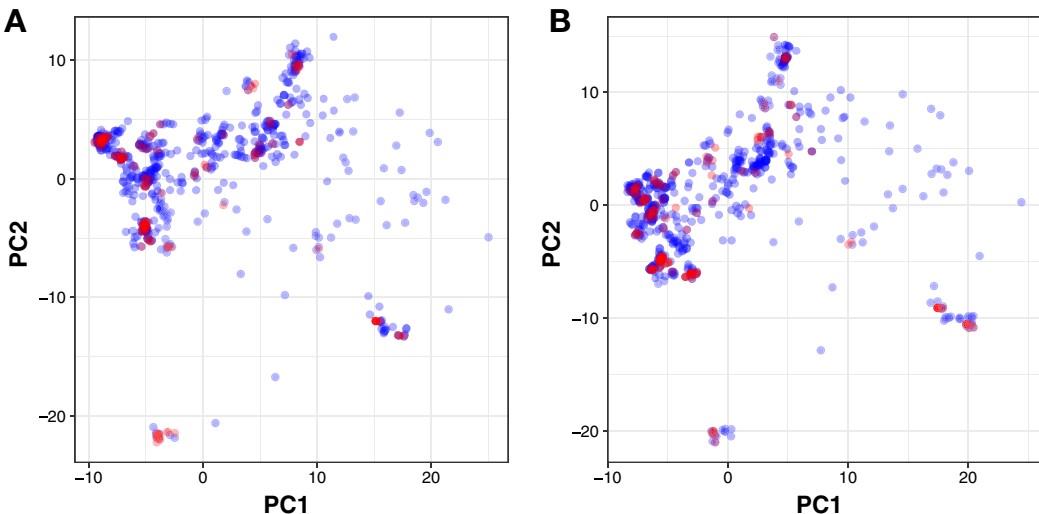

**Figure 5 PCA scores plot for compounds in the ERα and ERβ datasets.** The scores plot allows visualization of the distribution of compounds for internal (blue) and external (red) sets that constitutes the assessment of the applicability domain for ERα (A) and ERβ (B) datasets.

the number of nodes in each parent scaffold lineages and were also subjected to extensive analysis as will be discussed in paragraphs hereafter. Fig. 6 shows the schematic process of the scaffold analysis.

As mentioned above, further in-depth exploration of ERα and ERβ inhibitors led to the identification of top actives and inactives as shown in Figs. S1 and S2, respectively. The top active compounds of ERα had a bioactivity range of 8.398–8.698 pIC$_{50}$ while the top active compounds of ERβ had a bioactivity range of 8.045–8.522 pIC$_{50}$. Likewise, the bioactivity range for the top inactive compounds pertaining to the ERα and ERβ groups were observed to have pIC$_{50}$ in the range of 3.008–3.252 and 3.000–3.587 pIC$_{50}$, respectively.

In addition, a rigorous analysis of the chemical space was conducted by investigating the underlying scaffold structures as presented in actives and inactives using the Scaffold Hunter software (*Schäfer et al., 2017*). Particularly, chemical structure clouds were created for actives and inactives of ERα and ERβ as shown in Fig. 7. Analysis of ERα actives led to the identification of two major scaffolds namely 2,3-dihydro-1,4-benzoxathiine (with a frequency of 74) and diaryltetrahydronaphthalene (with a frequency of 27). On the other hand, ERα inactives consisted of five major scaffolds namely 4-hydroxypyrimidine (with a frequency of 26), 3-phenyl-4-(3H)quinazolinone (having 4-hydroxypyrimidine as the parent with a frequency of 24), chromane (with a frequency of two), pyran-4-one (with a frequency of 8) and leucoline (with a frequency of 17). Similarly, analysis of ERβ actives led to the identification of five top scaffolds whereby one is in common with ERα actives (i.e, lasofoxifene having a frequency of three). The other 4 scaffolds identified included 2-phenyl-1H-inden-1-one (with a frequency of nine), 2-phenylnaphthalene (with a frequency of 59), 2-phenylbenzofuran (with a frequency of 21), and 1,2,9,9a-tetrahydrofluoren-3-one (with a frequency of 25). Furthermore, three major scaffolds were

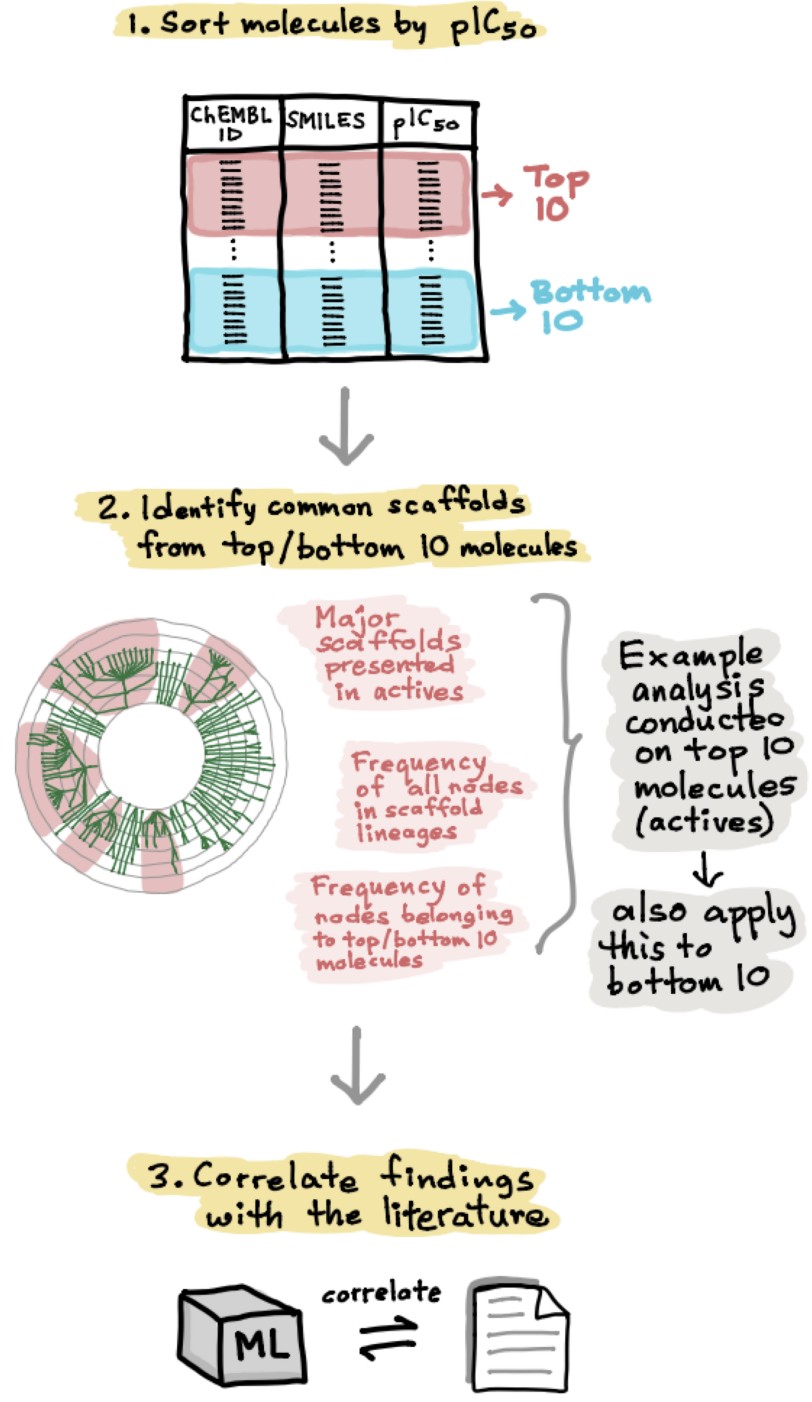

**Figure 6 Schematic representation of the methodological workflow of obtaining the scaffolds for ERα and ERβ active and inactive groups.** Top active and inactive compounds were determined from their pIC50 values. Scaffold Hunter was used to create scaffold trees whereby top scaffolds were determined for each bioactivity class (i.e., actives and inactives) for both ERα and ERβ.

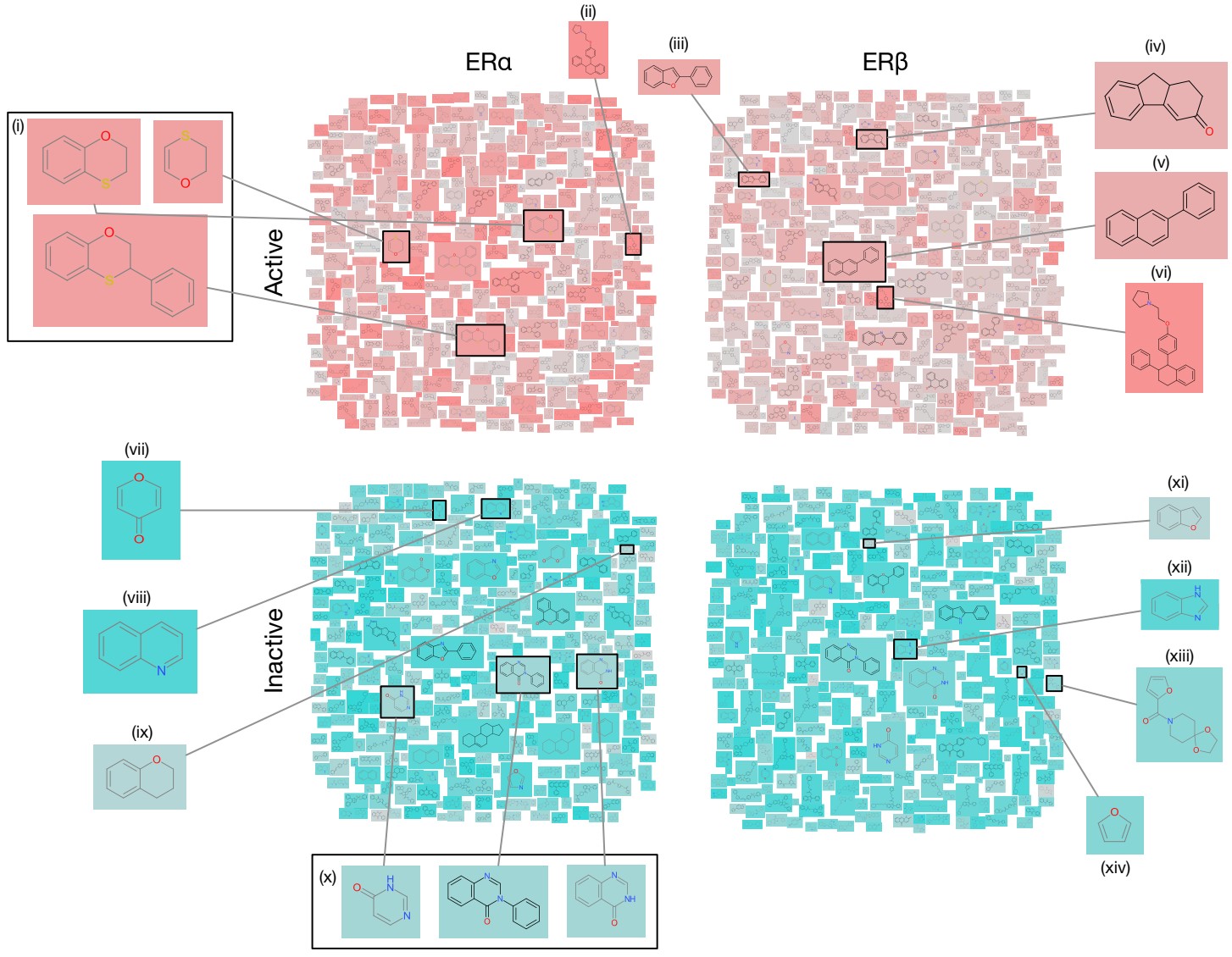

(i) 2,3-Dihydro-1,4-benzoxathiine  (ii) diaryltetrahydronaphthalene  (iii) 2-phenylbenzofuran  (iv) 1,2,9,9a-tetrahydrofluoren-3-one  (v) 2-phenylnaphthalene  (vi) diaryltetrahydronaphthalene

(vii) pyran-4-one  (viii) leucoline  (ix) chromane  (x) 4-hydroxypyrimidine and 3-phenyl-4-(3H)quinazolinone  (xi) coumarone  (xii) prazolopyrimidine

(xiii) 1,4-Dioxa-8-azaspiro[4.5]dec-8-yl(2-furyl)methanone  (xiv) furan

**Figure 7 Chemical structures cloud of ERα and ERβ actives and inactives.** Chemical structure cloud of actives (top panels) and inactives (bottom panels) for ERα and ERβ inhibitors. Particularly, active compounds were defined as compounds having IC50 in the range of 10–1,000 nM whereby highly actives (>10 nM) are represented by salmon pink color while weakly actives (<10 nM) are represented in grey. Inactive compounds were defined as compounds having IC50 in the range of 10,000–1,000,000 nM whereby the teal color corresponds to the most inactive compounds (towards the 1,000,000 nM scale) while the grey color corresponds to the higher inactive compounds (towards the 10,000 nM scale).

observed for ERβ inactives, which comprises of coumarone, pyrazolopyrimidine combined with a furan and 1,4-dioxa-8-azaspiro[4.5]dec-8-yl(2-furyl)methanone. The importance of these above-mentioned scaffolds will be discussed in the Structural interpretation section below.

**Table 1 List of 12 sets of fingerprint descriptors calculated from the PaDEL-Descriptor software.**

| Fingerprint | Number | Description |
| --- | --- | --- |
| 2D Atom Pairs | 780 | Presence/absence of atom pairs for various topological distances |
| 2D Atom Pairs Count | 780 | Frequency count of atom pairs for various topological distances |
| E-state | 79 | Electrotopological state atom types |
| CDK | 1,024 | Fingerprint of length 1,024 and search depth of 8 |
| CDK Extended | 1,024 | Extends the fingerprint with additional bits describing ring features |
| CDK Graph Only | 1,024 | A special version considering only the connectivity and not the bond order |
| Klekota-Roth | 4,860 | Presence/absence of SMARTS patterns for functional groups |
| Klekota-Roth Count | 4,860 | Frequency count of SMARTS patterns for functional groups |
| MACCS | 166 | Binary representation of chemical features defined by MACCS keys |
| PubChem | 881 | Binary representation of substructures as defined by PubChem |
| Substructure | 307 | Presence/absence of chemical substructures |
| Substructure Count | 307 | Frequency count of chemical substructures |

## QSAR modeling

This study follows the Organisation for Economic Co-operation and Development (OECD) (*OECD, 2014*) guidelines for the development of robust QSAR models. These guidelines are applied in all our work as previously mentioned (*Malik et al., 2020*) and comprises of the following main points: (i) the data set has a defined endpoint, (ii) uses an unambiguous learning algorithm, (iii) the applicability domain of the QSAR model is well defined, (iv) appropriate measures of goodness-of-fit, robustness and predictivity and (v) mechanistic interpretation of the QSAR model. Following these aforementioned guidelines to develop interpretable QSAR models, this study makes use of molecular fingerprints that are interpretable which are computed using the PaDEL-Descriptor software. As shown in Table 1, three out of the 12 fingerprints (i.e., PubChem, Substructure and Klekota–Roth) are readily interpretable. In addition, Table 2 provides details on the model performances for all 12 fingerprints.

In this study, we developed a QSAR model based on the random forest algorithm in order to differentiate the active and inactive inhibitors for ERα and ERβ subtypes. Table 2 shows the results from the RF model with 12 different types of fingerprints over an internal validation test, CV and an external validation test. The best averaged values were observed as Ac of 94.65% and 92.25% and MCC of 0.89 and 0.76 for ERα and ERβ, respectively which was achieved for the PubChem fingerprint descriptors as evaluated by CV. Concurrently, Klekota–Roth and Substructure descriptors also performed well harbouring the second and third highest averaged values for Ac and MCC in which models built using Klekota–Roth fingerprints afforded Ac and MCC values of 90.83% and 0.81%, respectively for ERα and Ac and MCC values of 94.36% and 0.82%, respectively for ERβ. Similarly, models built using the Substructure fingerprints afforded Ac and MCC values of 93.89% and 0.87%, respectively for ERα and Ac and MCC values of 94.36% and 0.82%, respectively for ERβ. Although the Ac and MCC values of models built using the PubChem fingerprints for ERβ were not superior to the models built using

**Table 2 Summary of model performance from classification models for ERα and ERβ.**

| Fingerprint | Training set | | | | 5-fold cross-validation | | | | Testing set | | | |
|---|---|---|---|---|---|---|---|---|---|---|---|---|
| | $Ac_{Tr}$ | $Sn_{Tr}$ | $Sp_{Tr}$ | $MCC_{Tr}$ | $Ac_{CV}$ | $Sn_{CV}$ | $Sp_{CV}$ | $MCC_{CV}$ | $Ac_{Test}$ | $Sn_{Test}$ | $Sp_{Test}$ | $MCC_{Test}$ |
| ERα | | | | | | | | | | | | |
| 2D Atom Pairs | 96.78 | 95.86 | 97.90 | 0.94 | 83.90 | 84.14 | 83.61 | 0.68 | 84.73 | 98.57 | 68.85 | 0.72 |
| 2D Atom Pairs Count | 100.00 | 100.00 | 100.00 | 1.00 | 87.31 | 87.50 | 87.08 | 0.74 | 97.71 | 98.57 | 96.72 | 0.95 |
| E-state | 89.01 | 89.47 | 88.48 | 0.78 | 83.33 | 84.95 | 81.53 | 0.67 | 90.08 | 94.29 | 85.25 | 0.80 |
| CDK | 96.59 | 96.82 | 96.33 | 0.93 | 88.83 | 87.84 | 90.09 | 0.78 | 93.89 | 100.00 | 86.89 | 0.88 |
| CDK Extended | 99.24 | 99.29 | 99.18 | 0.98 | 89.39 | 89.00 | 89.87 | 0.79 | 94.66 | 98.57 | 90.16 | 0.89 |
| CDK Graph Only | 97.34 | 97.19 | 97.53 | 0.95 | 86.93 | 86.64 | 87.29 | 0.74 | 86.26 | 98.57 | 72.13 | 0.74 |
| Klekota-Roth | 93.56 | 95.60 | 91.37 | 0.87 | 86.93 | 89.34 | 84.38 | 0.74 | 90.84 | 88.57 | 93.44 | 0.82 |
| Klekota-Roth Count | 95.83 | 98.15 | 93.39 | 0.92 | 88.83 | 91.18 | 86.33 | 0.78 | 86.26 | 81.43 | 91.80 | 0.73 |
| MACCS | 96.40 | 96.15 | 96.69 | 0.93 | 84.66 | 85.31 | 83.88 | 0.69 | 96.95 | 95.71 | 98.36 | 0.94 |
| PubChem | 96.40 | 96.15 | 96.69 | 0.93 | 87.69 | 86.09 | 89.82 | 0.75 | 94.66 | 100.00 | 88.52 | 0.90 |
| Substructure | 92.99 | 93.93 | 91.94 | 0.86 | 83.52 | 86.57 | 80.38 | 0.67 | 93.89 | 97.14 | 90.16 | 0.88 |
| Substructure Count | 91.28 | 93.09 | 89.33 | 0.83 | 82.95 | 85.87 | 79.92 | 0.66 | 93.89 | 97.14 | 90.16 | 0.88 |
| ERβ | | | | | | | | | | | | |
| 2D Atom Pairs | 96.68 | 96.32 | 98.18 | 0.90 | 86.01 | 87.53 | 77.11 | 0.55 | 88.73 | 99.10 | 51.61 | 0.65 |
| 2D Atom Pairs Count | 99.65 | 99.55 | 100.00 | 0.99 | 88.46 | 90.11 | 80.41 | 0.64 | 92.96 | 100.00 | 67.74 | 0.79 |
| E-state | 94.06 | 94.03 | 94.17 | 0.82 | 87.24 | 89.45 | 76.53 | 0.60 | 90.14 | 98.20 | 61.29 | 0.69 |
| CDK | 99.13 | 99.11 | 99.18 | 0.97 | 90.03 | 91.14 | 84.69 | 0.69 | 95.77 | 98.20 | 87.10 | 0.87 |
| CDK Extended | 98.95 | 98.89 | 99.17 | 0.97 | 90.73 | 92.09 | 84.62 | 0.72 | 94.37 | 99.10 | 77.42 | 0.83 |
| CDK Graph Only | 96.85 | 96.53 | 98.20 | 0.91 | 87.41 | 89.31 | 77.89 | 0.61 | 91.55 | 98.20 | 67.74 | 0.74 |
| Klekota-Roth | 98.25 | 98.23 | 98.32 | 0.95 | 89.16 | 90.87 | 81.19 | 0.66 | 94.37 | 99.10 | 77.42 | 0.83 |
| Klekota-Roth Count | 98.95 | 98.89 | 99.17 | 0.97 | 90.38 | 91.53 | 85.00 | 0.70 | 94.37 | 99.10 | 77.42 | 0.83 |
| MACCS | 99.48 | 99.78 | 98.41 | 0.98 | 88.81 | 91.18 | 78.50 | 0.66 | 95.07 | 99.10 | 80.65 | 0.85 |
| PubChem | 98.25 | 98.02 | 99.15 | 0.95 | 90.38 | 91.53 | 85.00 | 0.70 | 92.25 | 99.10 | 67.74 | 0.76 |
| Substructure | 95.10 | 94.67 | 97.09 | 0.85 | 87.94 | 89.71 | 79.17 | 0.62 | 94.37 | 99.10 | 77.42 | 0.83 |
| Substructure Count | 99.30 | 99.33 | 99.19 | 0.98 | 89.34 | 91.24 | 80.77 | 0.67 | 95.77 | 99.10 | 83.87 | 0.87 |

fingerprints of Klekota–Roth and Substructure, they are quite comparable. In addition, taking into account the Ac and MCC values as well as the overall external and CV, for both ERα and ERβ and the interpretability of the features, we determined that the PubChem fingerprints were the ideal choice for interpretation of the model.

## Mechanistic interpretation of feature importance

In order to provide a better understanding of the mechanistic details governing ERα and β subtypes, an analysis of the feature importance on selected informative descriptors was conducted. Owing to the inbuilt ability of feature importance estimation of the RF model and its great prediction performance, this study utilized it for analysis. Generally, two measures are used to rank the important features, namely the mean decrease of the Gini index and the mean decrease of the accuracy. As reported by *Calle & Urrea (2011)* that the Gini index had more robust results compared to those from the accuracy,

![PeerJ]

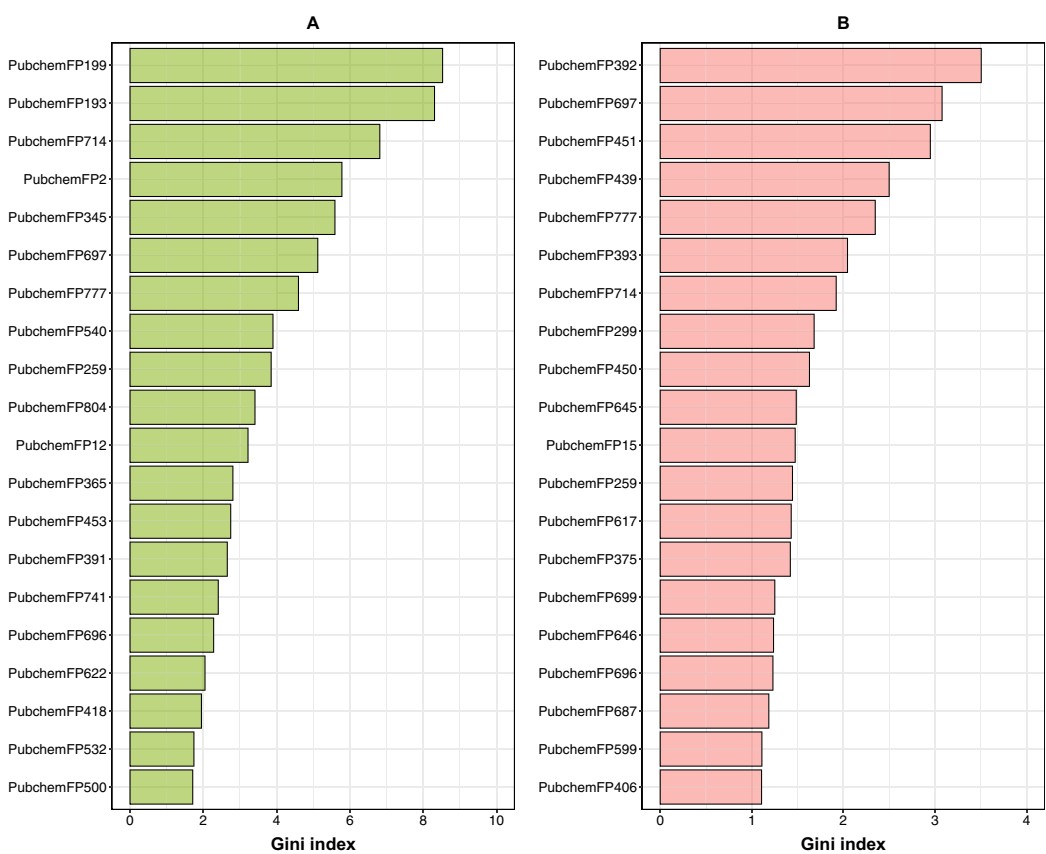

**Figure 8 Feature importance plot from ERα and ERβ models.** Box plots of the top 20 features as deduced from the Gini index from RF models built using PubChem fingerprints for both ERα (A) and ERβ (B).

we utilized the mean decrease of the Gini index to rank the importance of the PubChem feature descriptors. The 20 top-ranked PubChem descriptors deduced from the Gini index as derived from the RF model can be found in Fig. 8 and the contributions of their substructure towards the overall functioning of compounds as shown in Tables 3 and 4, will be discussed in the following section.

For ERα, the Gini index pertaining to the top 20 ranked features are shown in Fig. 8; and described in Table 3, which consisted of descriptors pertaining to the following classes: 7 aromatic (2 of which contain sulfur), 6 nitrogen containing features (consisting of amine and amide), 2 non-aromatic sulfur containing compound and 5 aliphatic hydrocarbons or atom counts. In addition, the Gini index pertaining to the top 20 ranked features for ERβ are shown in Fig. 8 and described in Table 4, which consisted of descriptors pertaining to the following classes: 10 nitrogen containing features (consisting of amine, amide and atom count), 3 aromatic, 3 alcohol, 4 aliphatic hydrocarbons or atom counts.

## Aromatic fingerprints

For ERα, the maximum number of PubChem fingerprints as obtained from the Gini index, with 7 out of 20 top-ranked features (i.e., PubChemFP199, PubChemFP193,

**Table 3 Summary of the top 20 features from the ERα model along with their corresponding SMARTS patterns and description.** The top features were obtained from the feature importance plot of the RF model.

| Features | SMARTS pattern | Substructure description |
|---|---|---|
| PubChemFP199 | >= 4 any ring size 6 | Greater than or equal to 4 six-membered cyclic ring |
| PubChemFP193 | >= 3 saturated or aromatic carbon-only ring size 6 | Greater than or equal to 3 saturated or aromatic carbon-only six-membered cyclic ring |
| PubChemFP714 | Cc1ccc(O)cc1 | 4-methylphenol |
| PubChemFP2 | >= 16 H | Greater than or equal to sixteen hydrogen atoms |
| PubChemFP345 | C(~C)(~H)(~N) | Ethylamine |
| PubChemFP697 | C-C-C-C-C-C(C)-C | 2-methylheptane |
| PubChemFP777 | CC1CCC(O)CC1 | 4-methylphenol |
| PubChemFP540 | C-N-C-[#1] | 1-(2-chloroethyl)-3-[2-[2-[[2-chloroethyl(nitroso)carbamoyl]amino]ethyldisulfanyl]ethyl]-1-nitrosourea |
| PubChemFP259 | >= 3 aromatic rings | Greater than or equal to 3 aromatic rings |
| PubChemFP804 | OC1CC(S)CCC1 | 3-sulfonyl phenol |
| PubChemFP12 | >= 16 C | Greater than or equal to sixteen carbon atoms |
| PubChemFP365 | C(~H)(~N) | Methanamine |
| PubChemFP453 | N(-C)(=C) | N-methylmethanimine |
| PubChemFP391 | N(~C)(~C)(~C) | N,N-dimethylmethanamine |
| PubChemFP741 | Oc1cc(S)ccc1 | 3-sulfonyl phenol |
| PubChemFP696 | C-C-C-C-C-C-C-C | Octane |
| PubChemFP622 | O=C-O-C:C | Ethyl formate |
| PubChemFP418 | C=N | Methanimine |
| PubChemFP532 | S-C:C-[#1] | Ethanethiol |
| PubChemFP500 | C-S-C:C | Methylsulfanylethane |

PubChemFP714, PubChemFP777, PubChemFP259, PubChem804 and PubChemFP741) was seen to pertain to the aromatic group. On the other hand, for ERβ, 3 out of the 20 top-ranked Gini features belonged to the aromatic group (i.e., PubChemFP777, PubChemFP714 and PubChemFP259). Surprisingly, all of the 3 aromatic Gini features for ERβ were also present for ERα. Therefore, we can infer that these 3 aromatic features are important to the functioning of the compounds. Diving deeper into the substructure description of these PubChem features, it can be seen that the first, second and ninth ranked features, (i.e., PubChemFP199, PubChemFP193 and PubChemFP259) correspond to aromatic rings of size ≥ 3 or 4. This point is in accordance to the fact that the natural agonist of ER (i.e. estradiol or E2) has only 1 aromatic ring while the most common antagonists of ER (i.e. tamoxifen and fulvestrant) have 3 and 1 aromatic rings, respectively (*Bafna et al., 2020*). In addition, it is the phenyl group in E2 that forms hydrogen bonds with Glu353 and Arg394 in the active site. Similarly, tamoxifen with its triphenylethylene core forms the same hydrogen bonds as E2 (*Bafna et al., 2020*). Thus, the phenyl moiety of compounds are important for ER inhibition. In addition, the third and seventh ranked features i.e., PubChemFP714 and PubChemFP777, correspond to 4-methylphenol which is an organic compound used as a precursor or intermediate for the manufacture of other chemicals. Furthermore, 4-methylphenol is also vital in the

**Table 4 Summary of top 20 features from the ERβ model along with their corresponding SMARTS patterns and description.** The top features were obtained from the feature importance plot of the RF model.

| Features | SMARTS pattern | Substructure description |
|---|---|---|
| PubChemFP392 | N(~C)(~C)(~H) | *N*-methylmethanamine |
| PubChemFP697 | C-C-C-C-C-C(C)-C | 2-methylheptane |
| PubChemFP451 | C(-N)(=O) | Formamide |
| PubChemFP439 | C(-C)(-N)(=O) | Acetamide |
| PubChemFP777 | CC1CCC(O)CC1 | 4-methylphenol |
| PubChemFP393 | N(~C)(~H) | Methanamine |
| PubChemFP714 | Cc1ccc(O)cc1 | 4-methylphenol |
| PubChemFP299 | N-H | Lambda1-azane |
| PubChemFP450 | C(-N)(=N) | Methanimidamide |
| PubChemFP645 | O=C-N-C-C | *N*-ethylformamide |
| PubChemFP15 | >= 2 N | Greater than or equal to two nitrogen atoms |
| PubChemFP259 | >= 3 aromatic rings | Greater than or equal to three aromatic rings |
| PubChemFP617 | C-C-C-O-[#1] | Propan-1-ol |
| PubChemFP375 | C(~N)(~N) | Methanediamine |
| PubChemFP699 | O-C-C-C-C-C(C)-C | 5-methylhexan-1-ol |
| PubChemFP646 | O=C-N-C-[#1] | *N*-methylformamide |
| PubChemFP696 | C-C-C-C-C-C-C-C | Octane |
| PubChemFP687 | O=C-C-C-C=O | Butanedial |
| PubChemFP599 | [#1]-C-C=C-[#1] | Prop-1-ene |
| PubChemFP406 | O(~C)(~H) | Methanol |

production of butylated hydroxytoluene (BHT) undergoing coupling to give an extensive family of diphenol antioxidants. These antioxidants are valued because they are relatively low in toxicity (*Fiege, 2000*). Although BHT has also been postulated as an antiviral drug, it has not yet been approved by any drug regulatory agency for use as an antiviral (*Pirtle, Sacks & Nachman, 1986*; *Lanigan & Yamarik, 2002*).

Furthermore, the tenth and fifteenth ranked features (i.e. PubChemFP804 and PubChemFP741) correspond to 3-sulfonyl phenol which according to the SMILES from its substructure description, seems to fit as a part of 4,4′-sulfonyldiphenol (Bisphenol S) (*National Center for Biotechnology Information, 2020*). As described in the report, Bisphenol S is an organic compound that has many functions, one of them being to act as an endocrine disruptor and it could thus modulate hormone receptors such as ERα and ERβ (*Rochester & Bolden, 2015*). For example, Viñas and Watson (*Viñas & Watson, 2013*) studied the nongenomic effects of Bisphenol S since it is an analogue of Bisphenol A, a well-known endocrine disruptor that imperfectly mimics the effects of physiologic estrogens via membrane-bound estrogen receptors. The authors concluded that Bisphenol S disrupts E2-induced cell signaling, leading to altered cell proliferation and cell death. Hence, the presence of such features in the Gini index top 20 is valid.

## Nitrogen-containing fingerprints

According to the Gini index for ERβ, 10 out of the top 20 features pertain to the nitrogen containing class which includes PubChemFP392, PubChemFP451, PubChemFP439, PubChemFP393, PubChemFP299, PubChemFP450, PubChemFP645, PubChemFP15, PubChemFP375 and PubChemFP646. On the other hand, features containing nitrogen as obtained from the Gini index for ERα constituted of 6 features, namely, PubChemFP345, PubChemFP540, PubChemFP365, PubChemFP453, PubChemFP391 and PubChemFP418. Taken together, features with nitrogen (i.e., amines and amides) constituted the maximum number of features, spanning both ER subtypes. Interestingly, there were no overlapping features found between the two groups for this category. Furthermore, the substitution of the CH group with a N atom in compounds containing aromatic and heteroaromatic ring systems, is a common bioisosteric transformation conducted to mimic the binding of natural ligands while exerting antagonistic effects (*Kumar et al., 2011*). Of note, six out of the 10 nitrogen containing features for ERβ (Table 4) belonged to the top 10 of Gini index, emphasizing their importance. In addition, most of the aforementioned features pertain to methanamine, *N*-methylmethanamine, ethylamine, *N*,*N*-dimethylmethanamine etc. which are all precursors of many significant chemical compounds such as Tamoxifen, 4-hydroxy tamoxifen, Raloxifene and many of their derived analogues. A review by *Sharma, Kumar & Narasimhan (2018)* highlights the substructures of ERα antagonists and their analogues which were analyzed in silico using molecular docking. Through this review, the authors emphasized the need for selective estrogen receptor antagonists for the treatment of breast cancer.

## Aliphatic hydrocarbons

From the analysis of the Gini index, for ERα and ERβ, respectively, 2 and 4 features out of the top 20 ranked features belonged to this group. PubChemFP697 and PubChemFP696 corresponding to 2-methlyheptane (which is also isomeric to octane) and octane, respectively, were observed to be a common feature for both ER subtypes (Tables 3 and 4). In addition, *Shoda et al. (2015)* observed that long alkyl side chains of tamoxifen derivatives acted as inducers of ER degradation. Furthermore, the same group designed a series of diphenylalkane derivatives bearing several long alkyl chains on the hydroxyl group and evaluated their biological properties such as ER degradation, binding affinity, transcriptional activity and anti-proliferation activity (*Shoda et al., 2015*; *Misawa et al., 2017*). Among all the compounds, one emerged as a novel ERα downregulator with a binding affinity of $IC_{50}$ = 4.9 nM and ERα antagonistic activity of $IC_{50}$ = 45 nM. Moreover, upon conducting computational docking analysis of the novel compound, the authors observed the interactions between the hydrogen atom on the amino group of the compound and the carboxylic acid of Glu351 of ERα which leads to the binding of the long alkyl chain to the hydrophobic groove of ERα. Thus, the amino group and the optimal length of the long alkyl chain in the diphenylheptane skeleton are considered important for ERα downregulation (*Nanjyo et al., 2019*).

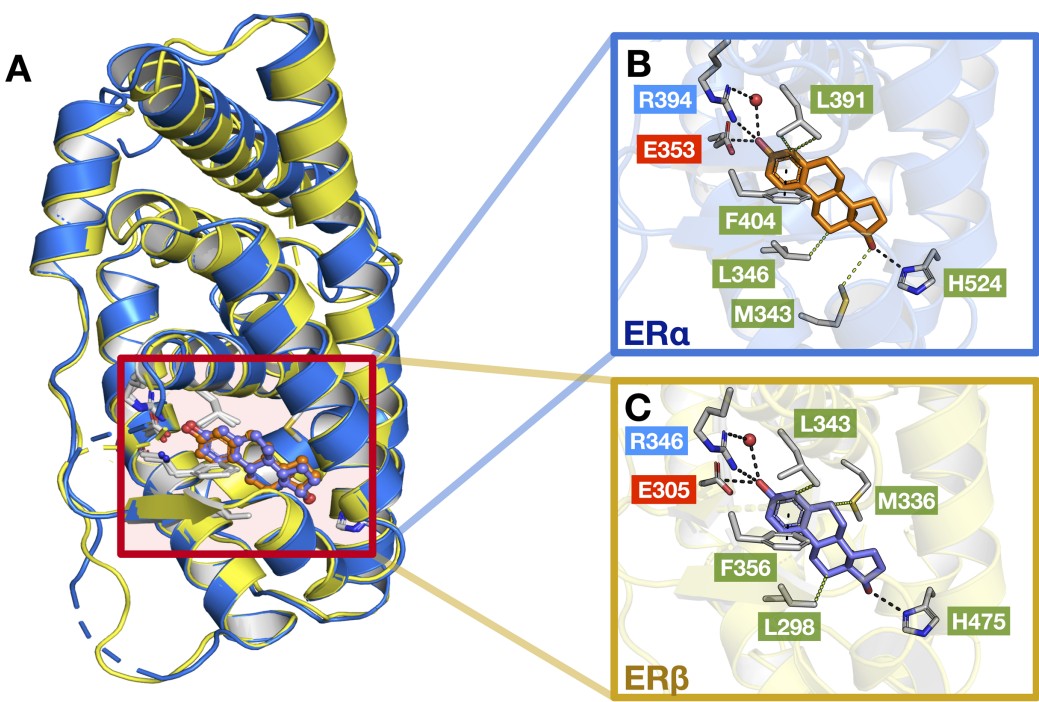

**Figure 9** **Protein structures of the two ER subtypes.** Superimposed structures of ERα (blue) and ERβ (yellow) bound to the E2 ligand (A). Close-up views of the binding cavity of ERα (B) and ERβ (C). Hydrophobic, negatively-charged and positively-charged residues are shown in green, red and blue colored text boxes, respectively.

## Structural interpretation

As previously mentioned, the LBD of ERα and ERβ show a consensus of around 55% and are composed of 12 α helices. Upon interaction with its natural ligand (i.e., E2), both ER subtypes form hydrogen bonds with residues (ERα/β numbering) Glu353/305, His524/475, Arg394/346 and a water bridge for Arg394/346 which connects to the A-ring of E2 (*Salentin et al., 2015*). In addition, both ERs also form perpendicular pi-stacking with the phenol rings of residue Phe404/356 and E2. Similarly, hydrophobic interactions were also observed with the interactions of both ER subtypes to E2 (consisting of residues M343, Leu346, Leu349, Ala350, Leu387 and Leu391 for ERα and Leu298, Leu301, Ala302, M336, Leu339, Leu343 and Leu476 for ERβ) however, ERβ possesses an additional interaction with residue Leu476 as compared to ERα (*Salentin et al., 2015*). Fig. 9 shows these aforementioned interactions, although some interactions are not shown for simplicity.

Generally, a good ER antagonist should possess two OH groups that are linked by a lipophilic central scaffold, which places them at a distance of about 11 Å. At least one of these hydroxyls should be a phenol or a phenol-bioisostere (i.e. replacement of the phenol group with another group that can act as a similar hydrogen bond donor) (*Sessler et al., 2017*). One of these OH groups form a strong hydrogen bond with residues (ERα/β numbering) Glu353/305, Arg394/346 and a water molecule and the other OH group, instead, mimics the OH group of estradiol and forms an additional hydrogen bond with

residue His524/475 in the estradiol D-ring pocket. At the other end of the binding cavity, the D-ring makes non-polar contacts with Ile424/376, Gly521/472 and Leu525/476 (*Brzozowski et al., 1997*). Beyond these key, energetically important interactions, which appear to be comparable between the two ERs, subtype-selectivity needs to arise from the different shape and hydrophobicity of the central scaffold. This has previously been demonstrated by *Pike et al. (1999)*, where the authors study the interactions of ERβ-Genistein (a ERβ-specific partial agonist) complex and observed that the H12 helix does not adopt the distinctive 'agonist' conformation but instead, lies in the 'antagonistic' position. In addition, it was further observed that two amino acid changes within the binding cavity (i.e., residues Leu384/Met336, and Met421/Ile373) may be responsible for specificity towards ERβ as well as differences in the propensity of the two ERs to form pockets of different sizes. In any case, the selective activation of ERα or ERβ may depend not only on a selective receptor binding affinity, but also on selective activation of each receptor subtype (*Leitman et al., 2010*).

Several studies have reported a novel class of Selective Estrogen Receptor Modulators (SERMs) based on dihydrobenzoxathiine (i.e., the top scaffold for ERα actives) as shown in Fig. 7 (*Kim et al., 2004*; *Chen et al., 2004*; *Viglianisi & Menichetti, 2010*). A SAR evaluation of several analogs was carried out for identifying the best structural features for SERM both in vitro and in vivo. It appears that the sulfur atom of the dihydrobenzoxathiine scaffold interacts with discriminative residues (Leu384 for ERα and Met336 for ERβ) in the binding pocket of the two receptor isoforms and thus plays a crucial role in maintaining the subtype-selectivity. This particular scaffold was observed in 5 of the top 10 active compounds (i.e., compounds **4**, **7**, **8**, **9** and **10** of Fig. S1) and therefore, its importance in preserving high activity towards ERα is evident. Another study revealed the diastereomerism of dihydrobenzoxathiine through molecular modeling (*Zhuang et al., 2011*) and discovered that the full antagonistic activity was achieved through hydrogen bonding with Glu353 and His524 of ERα LBD while van der Waals interactions were most predominant in the binding. In addition, lasofoxifene (derived from diaryltetrahydronaphthalene) was seen to be another major scaffold found in both ERα and ERβ active groups. Lasofoxifene selectively binds to both ERs with high affinity that is found to be similar to that of estradiol as well as other reported SERMs (i.e., raloxifene and tamoxifen) (*Gennari et al., 2006*). Furthermore, the crystal structure of lasofoxifene bound to ERα revealed features of ERα/SERM recognition whereby the C-terminal AF-2 helix is displaced at the LXXLL motif of coactivator proteins (occupying the space normally filled by residue Leu 540) as well as modulating the conformation of helix 11 residues (His 524, Leu 525) (*Vajdos et al., 2007*).

Additionally, the 2-phenylnaphthalene scaffold was seen to have a high frequency number for ERβ (Fig. S2). Genistein is a well known ERβ selective inhibitor that is derived from the 2-phenylnaphthalene scaffold. Furthermore, genistein is a key compound in the further identification of ERβ specific inhibitors (*Mewshaw et al., 2005*). The authors introduced substitutions at the appropriate positions of the phenylnaphthalene scaffold allowing a single orientation to predominate, which accounted for higher ERβ selectivity. Furthermore, *Wilkening et al. (2006)* noted that by creating analogues of

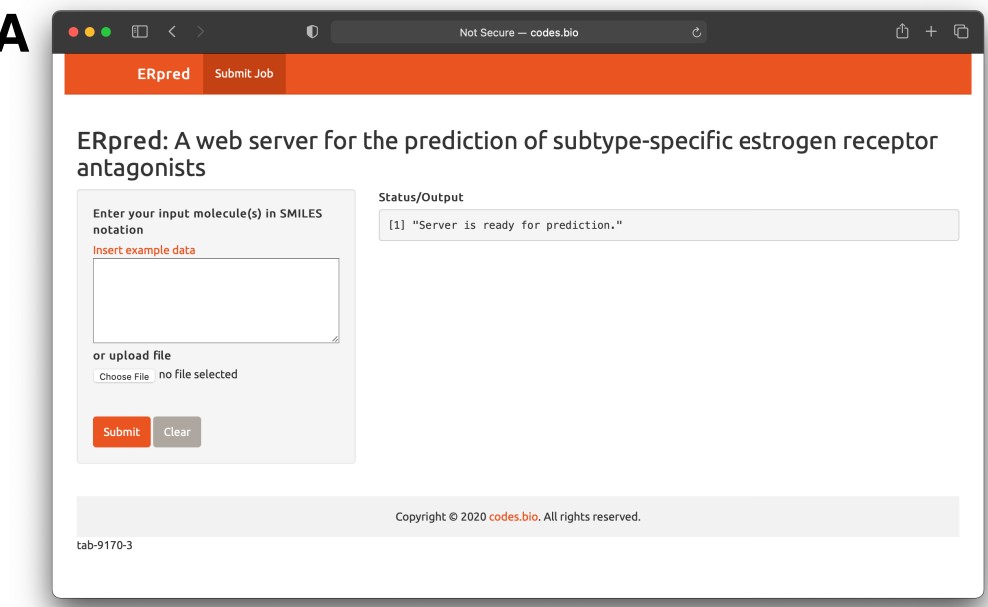

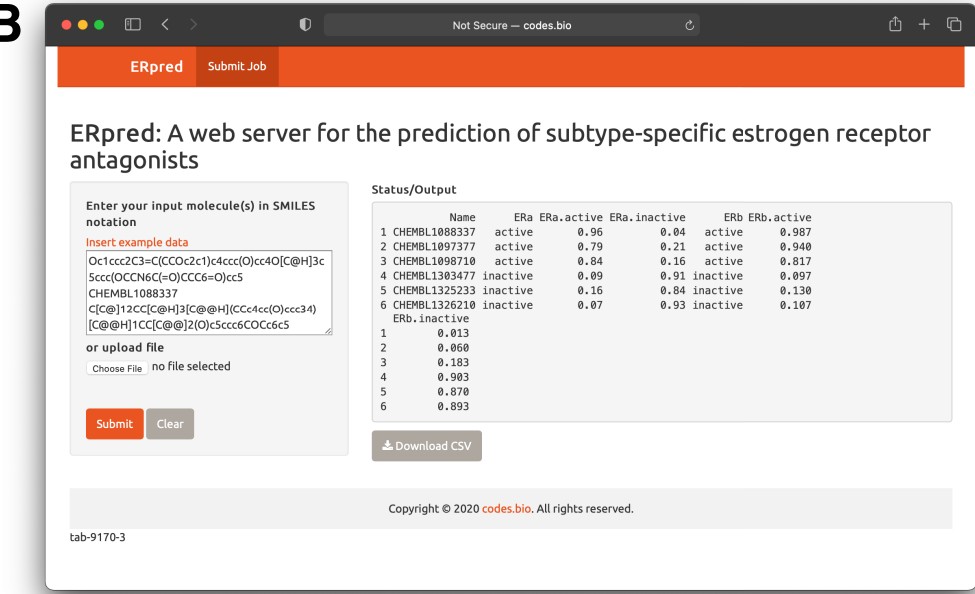

**Figure 10 Screenshots of the ERpred web server.** Upon loading of the web server a blank input box is shown (A) where users can enter or paste their SMILES notations for use as input for predictions to be made. After predictions are made, results are displayed under the "Status/Output" heading (B).

tetrahydrofluorenone through substitution, potent ERβ subtype selective ligands were formed. It should be noted that tetrahydrofluorenone accounted for 3 of the top 10 from ERβ actives (i.e., compounds **23**, **25** and **26**) as shown in Fig. S2. Furthermore, the authors had also reported several analogs possessing ERβ binding affinities that is

comparable to that of 17β-estradiol but with greater than 75-fold selectivity over that of ERα. Therefore, this analysis could help guide the process of novel ER inhibitor development that are subtype-selective.

## Model deployment as a web server

In order to make the prediction model presented herein a practical tool that can be widely used by the scientific community, we have constructed a web server called the ERpred using the model as described in previous sections. Briefly, the SMILES notation of the chemical compound of interest is used as the input which is fed into the ERpred web server. The server performs descriptor calculation using the PaDEL-Descriptor software and outputs predictions in the form of the bioactivity class label based upon the constructed random forest model. In addition, other web servers established for bioactivity prediction do so for just one class however, the ERpred web server is unique since it can predict the bioactivity for 2 proteins (herein ERα and ERβ), simultaneously. The Shiny package under the R programming environment was used to establish the web interface which has been made freely accessible at http://codes.bio/erpred/. Fig. 10 provides screenshots of the ERpred web server where the web server prior to input submission (i.e., panel A) and after prediction (i.e., panel B) are clearly depicted.

Briefly, a step-by-step guide on using the web server is given below:

- Step 1. The following URL (http://codes.bio/erpred/) should be entered into the web browser.
- Step 2. SMILES notation for common compounds of interest can be obtained from public databases such as ChEMBL, PubChem or ChemSpider whereas custom molecules can also be drawn into ChemDraw or ChemAxon MarvinSketch so as to generate the SMILES notation.
- Step 3. Input molecules as SMILES notation should be entered into the Input box or uploaded as a file containing the SMILES notation by clicking on the "Choose file" button.
- Step 4. Prediction process can be initiated upon clicking the "Submit" button.
- Step 5. Prediction results are automatically displayed in the grey box found below the "Status/Output" heading. Typically, it takes a few seconds for the server to process the task. Users can also download the prediction results as a CSV file by pressing on the "Download CSV button".

## CONCLUSIONS

Breast cancer is the most frequently detected cancer among women with over 2 million new cases and an estimated 627,000 deaths (15% of all cancer deaths in women) in 2018 (*World Health Organization, 2018*). However, current inhibitors and hormone therapy is problematic due to the development of resistance and an increased risk of breast and endometrial cancers, and thromboembolism. Thus, this study had qualitatively and quantitatively addressed these issues by building a QSAR model capable of distinguishing

active compounds from inactive compounds for ERα and ERβ. In addition, to allow researchers from all backgrounds easy access to our prediction model, we built a web server for discriminating compounds for ERα and ERβ with selectivity. ER activity prediction was evaluated via machine learning algorithms and several classes of fingerprint descriptors. The results obtained indicated that the RF algorithm coupled with the PubChem fingerprints allowed for the most interpretable descriptors along with the best performing model. The feature analysis of important substructure contributions as obtained from the Gini index revealed that aromaticity, amine groups and aliphatic hydrocarbons were important for the active compounds. Moreover, in-depth scaffold analysis of the top active compounds revealed that the binding specificity for ERα and ERβ involve different scaffolds. Since not many studies have focused on ERβ, this protein is worth further explorations. Thus, the knowledge gained from this study serves as general guidelines for the data driven design of potentially active and selective estrogen receptor antagonists.

### Funding
This work was supported by the New Researcher Grant (A31/2561) from Mahidol University as well as the TRF Research Career Development Grant (No. RSA6280075) from the Thailand Research Fund, the Office of Higher Education Commission and Mahidol University. The funders had no role in study design, data collection and analysis, decision to publish, or preparation of the manuscript.

### Grant Disclosures
The following grant information was disclosed by the authors:
Mahidol University: A31/2561.
Thailand Research Fund, the Office of Higher Education Commission and Mahidol University: RSA6280075.

### Competing Interests
The authors declare that they have no competing interests.

### Author Contributions
- Nalini Schaduangrat conceived and designed the experiments, performed the experiments, analyzed the data, prepared figures and/or tables, authored or reviewed drafts of the paper, and approved the final draft.
- Aijaz Ahmad Malik performed the experiments, analyzed the data, prepared figures and/or tables, and approved the final draft.
- Chanin Nantasenamat conceived and designed the experiments, performed the experiments, analyzed the data, authored or reviewed drafts of the paper, and approved the final draft.

# PeerJ

## Data Availability

The codes are available at GitHub: https://github.com/chaninlab/erpred.

The model is available at ERpred: http://codes.bio/erpred/.

## Supplemental Information

Supplemental information for this article can be found online at http://dx.doi.org/10.7717/peerj.11716#supplemental-information.

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
