# Peer review of "ERpred: a web server for the prediction of subtype-specific estrogen receptor antagonists"

_PeerJ, doi:10.7717/peerj.11716_

## Round 0.1 · original submission · Major Revisions

· Academic Editor

Major Revisions

The reviewers have expressed interest in this work and have found the study well conducted. However, they have also raised some concerns about the methodology and have asked for some clarifications and additional testing. I also recommend providing a guide and a few additional examples for the online tool, so its different usages can be demonstrated to a broader audience, especially to those who are new to the field.

Reviewer 1 ·

Basic reporting

In Keywords: QSAR is quantitative structure-activity relationship. No need to have both.

When the abbreviated names appear in the manuscript at the first time, the full names should be shown and the abbreviated comes after in parentheses. Line 30: MCC; Line 97: LDB. Full names should be shown here.

There are numerous grammar and language issues, which need to be addressed. For example: lines 19-20, 108-110, 159, 355, etc. Several long sentences are confusing and need to be simplified and rephrased, eg. Lines 101-104, 242-246.

Line 120, “8” should be deleted.

Line 134-135, format is wrong for the citation.

Line 214, should be Matthews correlation coefficient not Matthew’s correlation coefficient.

Experimental design

ERs are nuclear receptors, so their ligands usually are classified as agonists and antagonists. Inhibitors are for enzymes. To use antagonists rather than inhibitors in this manuscript is more scientifically sound.

As we all know, the quality of dataset is crucial for the QSAR modeling. We need to know more details about the data sources and information. The manuscript should explain what assays were performed to get IC50s in the original articles. In vitro binding and functional assays could give very different results, sometime more than 1 log unit, which will affect the classification of actives (pIC50 >=6) and inactives (pIC50>=5). Also, how the ligands were prepared before calculations of descriptors is not clear in the current manuscript. Details should be provided in Experimental Section, (eg, protonation, inclusion of racemic compounds, etc.) because different descriptors will be obtained for different ligand preparations. In addition, it is good to show what the structural diversity of the dataset is.

The authors used the Kennard Stone sampling method; however, the iterative random sampling method is more commonly used and generates more reliable results. Authors should try iterative random sampling and compare with Kennard Stone method to see how different the results are.

The current discussion in “Structural interpretation of differences in binding sites” section is irrelevant to the main topic of this manuscript. As the authored mentioned, the LBDs of ERα and ERβ share 55% sequence identity. Antagonists for both receptors should have cross binding affinity for the other receptor, and structurally may contain similar features for binding potency and different features for specificity, as QSAR studies show. In this section, authors are recommended to discuss how the similar and different features bind to the receptors and contribute to the potency and specificity structurally.

In the Chemical space analysis section, ALogP and MW were shown to be significant different between actives and inactives for ER-alpha. The means for both properties are not clear, and should be stated in the texts.

In line 110-113, it described what happens after binding to agonists to ERs. Since this manuscript focuses on antagonists, authors should add the description here about what will happen after antagonists’ binding to the receptor.

What is the percentage of actives and inactive in the internal and external dataset for both ER receptors? They may impact the performance of the QSAR models.

Line 368-369, PubChemFP777 is not 4-methylphenol, but hydrogenated one, the description about 4-methylphenol is irrelevant to the discussion

Line: 494-497: “Thus, this study aimed to qualitatively and quantitatively address these issues by building a QSAR model that would distinguish active compounds from inactive compounds for ERα and ERβ while also predicting the binding affinity for active compounds towards either ER subtype”. The current QSAR models are for classification of actives and inactives only. How can they predict binding affinity for actives?

Validity of the findings

no comment

Additional comments

This manuscript “ERpred: A web server for the prediction of estrogen receptor subtype-specific bioactive inhibitors” described the development of classification QSAR models for estrogen receptor (ER) antagonists by extracting ligands from the ChEMBL database and then a publicly available web server (ERpred) based on the models. ER is a member of the nuclear receptor family and consists of two main subtypes (ERα and ERβ), which have been shown to paly different roles in the development and metastasis of breast cancer. The development of compounds possessing ER subtype specificity still represents unmet medical need and warrants further studies. Publication of this manuscript will add values to the scientific community.

Reviewer 2 ·

Basic reporting

The manuscript of Schaduangrat and coworkers (#52348) introduces ERpred, a web server tailored for the prediction of inhibitors for the two estrogen receptor (ER) proteins alpha and beta, two well known pharmacological targets for breast cancer. By performing a classification structure-activity relationship (CSAR) of known inhibitors from the ChEMBL database, a set of structural features important for the compound activity was determined by using a random forest algorithm, which is a commonly used classification method. A number of molecular characteristics (e.g., presence of aromatic rings, or other functional groups) were identified, which are used in the web server to predict bioactivity of novel compounds towards the protein targets.

This work is interesting and, as far as I can see, well conducted. The results do not seem to be entirely outstanding, but predicting the existence of active and selective molecules for two proteins simultaneously is admittedly not an easy task. The ERpred web server could serve the scientific community by providing predictions that can help to filter out compounds in the pipeline of the selection of novel ER-targeting compounds. The manuscript is also generally well written, with a few minor mistakes (some of them will be pointed out below). Figures and tables are also clear and informative.

Some suggestions:

Line 20, “Estrogen receptor alpha and beta [...] represents the fundamental and critical determinants of clinical outcomes”. This part is not clear and should be changed. What is it supposed to be critical (their existence, abundance, or...)? To me, they are just pharmacological protein targets.

Line 29, “Ac of 94.65% 30 and 92.25% and MCC of 0.89 and 0.76 for ERa and ERß, respectively”. These quantities are not defined yet, and it seems also strange that they are not expressed with the same notation (0.89 = 89%).

Line 44: “Thailand, breast cancer ranks third behind lung and liver cancers accounting for an estimated 11.4% of new cases and 114,199 deaths in 2018 (International Agency for Research on Cancer, 2019)”. This is relevant for the Authors and their compatriots, but much less in the international context of the wide readership of this journal.

Line 54: “desperately needed”. I would use a less alarming adverb.

Line 56: “Estrogen receptors (ER) is a...”. Please correct this part.

Line 69: “more prominent role on the mammary gland and uterus”. Which type of role?

Line 83: “the effectiveness of this drug”. It is not entirely clear which compound is being referred to, and I would suggest to name it explicitly.

Line 134: “Estrogen receptor binding affinity estimated by QSAR”. This is cited as a reference, but citation is not clear.

Line 158: “molecular descriptors are important for QSAR studies as it characterize”. Please correct this part.

Line 169: “constant and near constant variables”. What do these quantities represent?

Line 206: “The mean decrease of the Gini index (MDGI) was utilized to estimate the important descriptors”. Maybe using a reference already here would be useful.

Line 212: “True positives (TP), true negatives (TN), false positives (FP) and false negatives (FN)”. Please correct or delete this sentence.

Line 219, Equation 2. What is the “=” intended for?

Line 233: “PCA bounding box”. This part is not clear. Please also note that, although it is pretty clear what PCA refers to, this acronym was never defined (and here it is used for the first time).

Line 269: “number of hydrogen bond donors (> 5)”. This should be “< 5”.

Line 282: “were observed to be less than those”. Maybe “lower”?

Line 379: “(Bisphenol S) (National Center for Biotechnology Information, 2020)”. This reference should be adequately explained, e.g. “as described in the report on...”.

Line 432: “water bridge for Arg394/346” (and similarly line 446, “water molecule...”). It is not entirely obvious to me what this refers to. Is it a crystallographic water bridging two Arg residues? Can a reference be given?

Line 436: “ERbeta possesses an additional binding (Leu476)”. Maybe “binder”? In any case, this expression is confusing.

Line 449: “appear to be commensurate between the two ERs”. The term “commensurate” is not clear.

Line 470: “the bioactivity for 2 classes of proteins (herein ERa and ERß)”. Why not simply “for two proteins”?

Line 501: “afforded the best performance while also affording...”. This part is not very elegant.

Line 499. “Looking forward, molecular dynamic simulations could play a role in analysis of ER subtype specificity which should be explored in future work”. This conclusion is really puzzling to me. It does not have anything to do with this work, and it is not clear how molecular dynamics (please note the ending “s” in “dynamics”) will be interesting as a continuation of this work. The sentence could be removed.

Experimental design

No comment.

Validity of the findings

No comment.

Additional comments
* * *
Reviewer 3 ·

Basic reporting

The authors have investigated a large-scale classification structure-activity relationship (CSAR) study of inhibitors from the ChEMBL database, which consisted of an initial set of 11,618 compounds. They are deployed as the publicly available web server called ERpred at http://codes.bio/erpred.
I have read the manuscript and put my comments wherever is needed. The manuscript is clearly written in professional, unambiguous language. However, the part of predicting the binding affinity for active compounds is not clear therefore more information for this part is needed.
The idea of the work is relevant and interesting and the article is publishable after making the following minor corrections:

Experimental design

'no comment'

Validity of the findings

'no comment'

Additional comments

1. Put the full name of ligand binding domains at lines 97.

2. Many references cited In Lines 133- 137: (Anstead, Carlson & Katzenellenbogen, 1997; Serafimova et al., 2007; Xiang et al., 2009; Toropov et al., 2012; Chang et al., 2013; “Estrogen receptor binding affinity estimated by QSAR,” 2015; Ribay et al., 2016; Suvannang et al., 2017, 2018; Lee & Barron, 2017; Pavlin et al., 2018; Pang et al., 2018; Balabin & Judson, 2018; Cotterill et al., 2019; Bafna et al., 2020). You must put the references in order of years and correct this reference “Estrogen receptor binding affinity estimated by QSAR,” 2015 (write it in the same style).

3. At line 302 put the full name of OECD

4. Step 2 is not quite clear (line 479- 481). Please explain this step

---

## Round 0.2 · accepted · Accept

· Academic Editor

Accept

Congratulations on addressing peer-review and editorial concerns and on acceptance of your paper. Please see the minor corrections suggested by the reviewer during the proof stage.

Reviewer 2 ·

Basic reporting
* * *
Experimental design
* * *
Validity of the findings
* * *
Additional comments

Minor suggestions:

Line 80: "liver and adipose tissues are mainly composed of ERalpha". The term "composed" is clearly excessive, something like "mainly include" is acceptable.

Line 346: "as showns". Typo.

Line 604: "as shown in Figure 7". It should be double-checked whether the old or new figure 7 is meant to be cited.